# Factors associated with reluctancy to acquire COVID-19 vaccination: A cross-sectional study in Shiraz, Iran, 2022

Najmeh Maharlouei[1], Parisa Hosseinpour[2], Amirhossein Erfani[3]*, Reza Shahriarirad[3], Hadi Raeisi Shahrakie[1], Abbas Rezaianzadeh[4,5], Kamran Bagheri Lankarani[1]

1 Health Policy Research Center, Institute of Health, Shiraz University of Medical Sciences, Shiraz, Iran, 2 School of Medicine, Islamic Azad University, Kazeroun Branch, Kazeroun, Iran, 3 Thoracic and Vascular Surgery Research Center, Shiraz University of Medical Sciences, Shiraz, Iran, 4 Colorectal Research Center, Shiraz University of Medical Sciences, Shiraz, Iran, 5 Non-Communicable Diseases Research Center, Shiraz University of Medical Sciences, Shiraz, Iran

* Ahnerfani@gmail.com

**Data Availability Statement:** All data regarding this study has been reported in the paper. Don't

## Abstract

### Background

Vaccination is a crucial action that can end the COVID-19 pandemic and reduce its detrimental effect on public health. Despite the availability of various vaccines, this study was conducted to better understand the factors behind individuals refusing to get vaccinated.

### Method

The current cross-sectional study was conducted with individuals above 18 years of age in Shiraz, Iran, who were eligible but refused to receive the COVID-19 vaccination. Demographic features and factors related to their hesitancy and willingness to participate in the vaccination program were recorded in a questionnaire.

### Result

Out of 801 participants in the current study, 427 (53.3%) were men, with a mean age of 37.92 years (± 14.16). The findings revealed that 350 (43.7%) participants claimed the side effects of the vaccine outweigh the benefits as one reason for their reluctance toward COVID-19 vaccination, followed by the unknown efficacy of vaccines (40.4%) and a lack of trust in vaccine companies (32.8%). Ensuring the safety of the vaccine (43.7%) and verifying its effectiveness (34.5%) were the most prevalent factors behind participating in the vaccination program. Those who reported their socio-economic status as low were significantly reluctant toward vaccination because of a self-presumption of high immunity ($p$-value < 0.001), the unclear efficacy of vaccines ($p$-value < 0.001), the side effects outweighing the benefits of vaccines ($p$-value < 0.001), distrust of vaccine companies ($p$-value < 0.001), usage of mask, gloves, and sanitizers ($p$-value < 0.001), contradictory speech of health authorities regarding vaccines ($p$-value = 0.041), and the unavailability of trusted vaccines ($p$-value = 0.002). It should also be noted that participants reported a greater likelihood to

hesitate to get in touch with the corresponding author in case further information is required.

**Funding:** Vice-chancellor for Research of Shiraz University of Medical Sciences financially supported this Study through Kamran Bagheri Lankarani (Grant No: 24974). The funders had no role in study design, data collection and analysis, decision to publish, or preparation of the manuscript.

**Competing interests:** The authors have declared that no competing interests exist.

**Abbreviations:** COVID-19, Coronavirus disease of 2019; WHO, World Health Organization; FDA, Food and Drug Administration.

obtain information about vaccination reluctance from family and friends (*p*-value <0.001) and complementary medicine professionals (*p*-value <0.001).

## Conclusion

Avoiding vaccination is an undeniable public and individual health concern in Iran, as demonstrated in the current study. Concern about vaccine efficacy and side effects is the most reported cause of vaccination reluctance among individuals, which could be altered by emphasizing mass education and averting an infodemic by forming dedicated multidisciplinary organizations.

## Introduction

In December 2019, an outbreak of coronavirus disease 2019 (COVID-19) [1, 2] was identified in Wuhan, China [3]. Despite attempts to contain the virus, the World Health Organization (WHO) soon declared a pandemic on the 11[th] of March 2020. Since then, more than 450 million people have contracted the disease and more than 6 million sufferers have died [4]. Alongside its effect on public health, the pandemic has triggered an enormous disruption in aspects of the economy and social life of people living in developing countries who are already dealing with difficulties, such as Iran [5, 6]. Thus, health authorities were obligated to take necessary actions to stop the spread of the virus to reduce its disturbing effects on public health.

Since the beginning of the pandemic, several tactics have been practiced to reduce the detrimental consequences of a pandemic, such as applying different available medicines [7], practicing lockdowns, social distancing, utilizing hand sanitizers, closure of public places, and travel restrictions [8]. Although these approaches helped flatten the pandemic curve, they were not the ultimate solution [9]. Vaccination is a crucial action to end the COVID-19 pandemic and reduce hospitalization and mortality [10]. However, the impact of vaccines on the pandemic also depends on several factors, such as the effectiveness of the vaccines, how quickly they are delivered, and how many people get vaccinated [10, 11]. Pfizer-BioNTech, Moderna's mRNA-1273, and AstraZeneca/Oxford's AZD1222 were among the first developed vaccines to obtain Food and Drug Administration (FDA) [12] approval for emergency use [13, 14]. Because of their remarkable effectiveness in reducing mortality and morbidity [15], the administration of these vaccines to most of the population was considered the only option for exiting the pandemic crisis. Although the percentage might differ among viruses due to the route of spread, the more people receive vaccination, the closer the population gets to herd immunity against COVID-19 [16].

While reaching a higher percentage of vaccination equals less morbidity and mortality, achieving this goal is not an easy task. Many countries face economic, cultural, and political challenges. To vaccinate sufficient numbers of people, vaccines must be available, convenient, and affordable. Nevertheless, despite the availability of vaccines, some people are reluctant or hesitant to be vaccinated [17, 18]. Previous findings showed that different factors could lead to COVID-19 vaccine hesitancy. Public distrust in the vaccine and its efficiency, lack of trust, belief that the vaccine has a political and harmful nature, and concerns about its safety are among the critical factors [19–21]. In addition, the overwhelming amount of misinformation on social media has made it harder for people to trust the vaccine [22, 23]. In many countries, vaccine hesitancy is high enough to endanger community immunity [18].

Iran (and other countries worldwide) began its vaccination program against COVID-19 on the 9th of February 2021 and accelerated it in September 2021 until 76.47% coverage (at least one dose) was achieved on the 12[th] of September 2022 [24]. The program was designed so that vaccination could be available for all of the population despite challenges. Vaccines were free and could be administered on-site to people who could not show up to vaccination centers or to nomads living far from the reach of health centers through mobile health managers. Those who did not show up for vaccination were called to be reminded of where and how they could receive the vaccines. Because it is crucial to understand the determinants of rejecting the vaccine despite its availability throughout the country while more vaccination equals less mortality and morbidity, this cross-sectional study was conducted in the fifth largest city of Iran. The findings of the current study will help local and global health authorities take the necessary actions for achieving healthier communities.

## Methods and materials

This cross-sectional study was conducted between March 8[th], 2022, and April 15[th], 2022, to investigate the factors associated with reluctance toward COVID-19 vaccination among those who had already been called in Shiraz, the fifth most populous city in the south-west of Iran.

### Sample size

The sample size was calculated to be 800 with a 95% confidence interval, 0.5% margin of error, 25% expected agreement, and the expected population size of 629,115 out of 3,583,549 who were eligible for vaccination as provided by the Vice-chancellor for Health.

### Sampling method

Personal and contact information for prospective participants was obtained with the help of the Vice-chancellor for Health, affiliated with Shiraz University of Medical Sciences (SUMS); the individuals were contacted by phone and requested to complete the relevant questionnaire until a sample size of 800 individuals was attained.

### Ethics approval and consent to participate

The Medical Ethics Committee of Shiraz University of Medical Sciences approved the present study with the code number IR.SUMS.REC.1400.814. The purpose of this study was thoroughly explained to the participants, and they were assured that their information would be kept confidential by the researchers. Verbal consent was also obtained from the participants.

### Inclusion and exclusion criteria

Inclusion criteria included being above 18 years of age, registered as inhabitants of Shiraz in the Fars Civil Registry Office, and not having referred for vaccination against COVID-19, even after having been called. Individuals who did not answer their phone after attempts were made on three different weekdays at various hours of the day or did not consent to participate in the survey and those who had been vaccinated in other countries were excluded from this study.

### Data gathering tool

The questionnaire used in the current study was designed based on a literature review and expert opinion and consisted of five parts: 1. Demographic features of participants and self-reported socio-economic status. 2. A self-rated health score between 0–10 (0 equaling poor health and 10 equaling excellent health) and a score of presumed susceptibility to COVID-19

infection between 0–10 (0 equaling low susceptibility and 10 equaling high susceptibility) 3. Questions regarding their reluctance toward vaccination against COVID-19 and motivation to participate in the vaccination program. 4. Source of information on reluctance to vaccinate. 5. The methods used by participants to prevent COVID-19 contraction.

## Study area

The current study was conducted in Shiraz, the capital of Fars province, located in southern Iran. Shiraz's 2022 metropolitan population was estimated to be 1,700,000, making it the fifth most populated city in Iran. With 41 hospitals and nearly 100 local health centers delivering primary health care, Shiraz is considered a center for health service delivery in southern Iran. Since February 2021, all local health centers throughout the metropolitan area of the city have offered vaccination with different types of vaccines, including domestic (COVIran Barekat, SpikoGen, PastoCoVac, RaziCovPars) and non-domestic ones (Sinopharm, Sputnik, AstraZeneca, COVAXIN). As of August 2022, all people above 18 years of age can receive their fourth dose of the vaccine in listed health centers.

## Data analysis

Descriptive statistics such as mean, standard deviation, and frequency distribution tables will be applied to describe the results. The chi-square or Fisher's exact test was used to compare the existing data, and $p$-values of less than 0.05 were considered significant. All data was analyzed using Statistical Package for Social Sciences (SPSS Inc., Chicago, Illinois, USA) version 26.0 software.

## Results

Out of 801 participants in the current study, 427 (53.3%) were men, and 502 (62.7%) were married. Mean participant age was 37.92 years (± 14.16), and participants had a mean number of 2.27 (± 1.69) children. Most of the participants were of Iranian nationality (95.4%), Fars ethnicity (74.8%), had a high school diploma (38.3%), and without supplementary insurance coverage (51.8%). The mean score of self-rated health was reported as 8.40 out of 10 (±1.93), and the mean score of presumed susceptibility to COVID-19 infection was reported as 4.64 out of 10 (± 2.80) (Table 1).

In the current study, 302 participants (37.7%) reported at least one incident of COVID-19 contraction, which led to the hospitalization of 15 (5%) individuals. In the majority of cases, COVID-19 contraction was diagnosed by a physician (181; 22.5%) or confirmed by Polymerase Chain Reaction (PCR) test (83; 10.3%). Fig 1.

As shown in Fig 2, the primary sources of information regarding the COVID-19 vaccine were family members and close friends (439; 54.8%), social media including Instagram, WhatsApp, Twitter, and Facebook (231, 28.8%), and national broadcasting (111, 13.9%).

The current findings revealed that 350 (43.7%) participants claimed the side effects of the vaccine outweighed the benefits as one of their reasons for reluctance toward COVID-19 vaccination, followed by the unknown efficacy of vaccines (40.4%) and a lack of trust in vaccine companies (32.8%). The reasons for vaccine reluctance are summarized in Table 2.

When asked under what conditions would they participate in the vaccination program, participants reported assurance of the safety of the vaccine (43.7%) and assurance of the effectiveness of the vaccine (34.5%) most frequently. An additional 204 (25.1%) participants claimed they would not get vaccinated under any circumstances (Table 3).

Fig 3 demonstrates the methods used by participants who were reluctant to vaccination to prevent COVID-19 contraction.

**Table 1. Demographic characteristics of those reluctant toward vaccination in Shiraz city in 2022.**

| Variable | Subgroup | Frequency | Percent (%) * |
|---|---|---|---|
| **Gender** | Male | 427 | 53.3 |
| | Female | 374 | 46.7 |
| **Marital status** | Single | 258 | 32.0 |
| | Married | 502 | 62.7 |
| | Widowed | 25 | 3.1 |
| | Divorced | 16 | 2 |
| **Highest education attainment** | Illiterate | 82 | 10.3 |
| | Below high school diploma | 302 | 37.8 |
| | High school diploma | 306 | 38.3 |
| | University degree | 110 | 13.8 |
| **Nationality** | Iranian | 764 | 95.4 |
| | Non-Iranian | 37 | 4.6 |
| **Ethnicity** | Fars | 591 | 74.8 |
| | Turk | 95 | 12 |
| | Lor | 65 | 8.2 |
| | Others | 39 | 4.8 |
| **Supplementary insurance coverage** | No | 415 | 51.8 |
| | Yes | 386 | 48.2 |
| **Socio-economic status** | Middle to high | 352 | 43.9 |
| | Middle to low | 92 | 11.5 |
| | Low | 357 | 44.6 |

*Percentage = frequency/801*100

As can be seen in Table 4, the male gender was significantly associated with reluctancy toward vaccination due to self-presumption of high immunity ($p$-value < 0.001), use of mask, gloves, and sanitizers ($p$-value = 0.021), disbelief in the existence of COVID-19 ($p$-value = 0.003), unavailability of the desired vaccine ($p$-value = 0.006), and belief that COVID-19 is not as severe as broadcasted ($p$-value = 0.010).

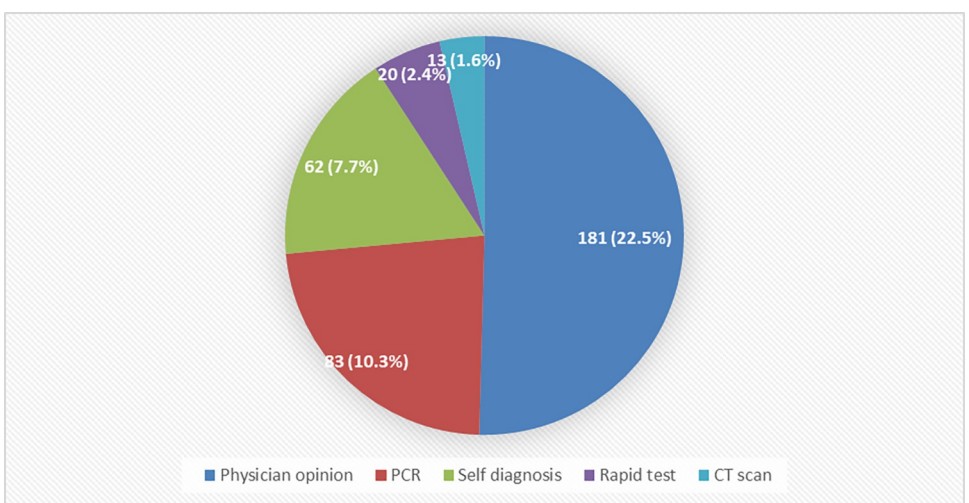

**Fig 1. Methods used to diagnose COVID-19 in 301 participants who reported at least one COVID-19 infection.**
PCR: Polymerase Chain Reaction; CT scan: Computed tomography scan.

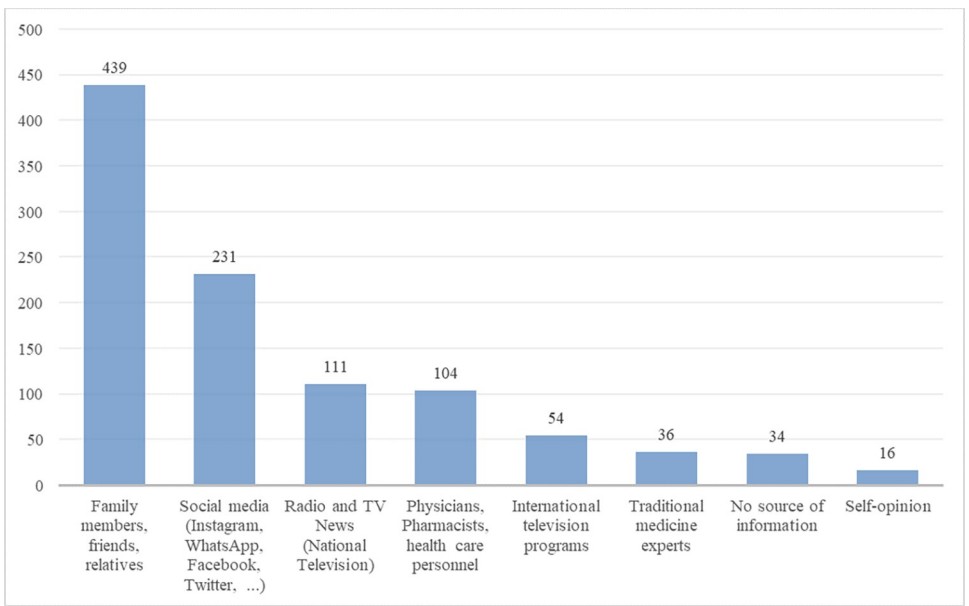

**Fig 2. Primary sources of information about the COVID-19 vaccine by the number of respondents.**

The female gender was significantly associated with future participation in the vaccination program if assurance of the common side effects of vaccines was provided ($p$-value = 0.012). In contrast, the male gender was significantly associated with never participating in vaccination under any circumstances ($p$-value < 0.001).

**Table 2. Reasons for reluctance toward COVID-19 vaccination.**

| Reasons for reluctance | Frequency; N = 801 | Percent (%) |
|---|---|---|
| The side effects of the vaccine outweigh the benefits. | 350 | 43.7 |
| The efficacy of vaccines is unknown. | 324 | 40.4 |
| I do not trust vaccine companies. | 263 | 32.8 |
| My body has a robust immune system. | 190 | 23.7 |
| I do not need to get the vaccine because of mask and glove use. | 105 | 13.1 |
| Life is in the hands of God, and there is no need for a vaccine. | 35 | 4.4 |
| I refuse due to pregnancy or lactation. | 34 | 4.2 |
| The vaccine I trust is not available. | 31 | 3.9 |
| I have no time to get vaccinated. | 30 | 3.7 |
| Health authorities' speech about vaccines is contradictory. | 28 | 3.5 |
| I have experienced reactions to previous vaccines. | 28 | 3.5 |
| I do not believe in the existence of COVID-19. | 27 | 3.4 |
| COVID-19 is not as intense as broadcasted. | 27 | 3.4 |
| I have already contracted COVID-19, so I do not need vaccination. | 26 | 3.2 |
| I trust the anti-vaxxers. | 20 | 2.5 |
| My decision is based on my physician's opinion. | 15 | 1.9 |
| I have a medical condition. | 13 | 1.6 |
| I believe in a conspiracy theory. | 6 | 0.7 |
| Vaccination contradicts my religious beliefs. | 3 | 0.4 |
| Other | 265 | 33.0 |

**Table 3. Response rate to the question "Under what conditions will you participate in the vaccination program?" among the unvaccinated population in Shiraz, Iran.**

| Reasons to participate in the future. | Frequency; N = 801 | Percent (%) |
|---|---|---|
| I am assured of the safety of the vaccine. | 350 | 43.7 |
| I am assured of the effectiveness of the vaccine. | 276 | 34.5 |
| I will not get vaccinated under any circumstances. | 204 | 25.5 |
| I need access to the desired vaccine. | 90 | 11.2 |
| I need secure access to approved non-domestic vaccines. | 39 | 4.9 |
| I need secure access to approved domestic vaccines | 19 | 2.4 |
| I will vaccinate after pregnancy and lactation | 21 | 2.6 |
| I will vaccinate if my physician advises me thus. | 14 | 1.7 |
| Others | 154 | 19.2 |

Aged between 18 and 25 years was significantly associated with reluctance toward vaccination because of self-presumed high immunity ($p$-value = 0.037) and trust of anti-vaxxers ($p$-value = 0.022). Furthermore, participants above 65 years of age believed that they should accept their destiny and thus felt that vaccination was not required ($p$-value = 0.036).

The current results revealed that female participants tended to get information about the COVID-19 vaccine from healthcare workers ($p$-value < 0.001). Male participants, however, tended to be informed through social media ($p$-value < 0.001) and radio and TV ($p$-value < 0.001) and believed in their own opinion rather than those of others ($p$-value = 0.006).

As shown in Table 5, participants with university degrees were significantly more reluctant toward vaccination because of contradiction in health authorities' speech about vaccination ($p$-value = 0.014); they were willing to participate in a vaccination program if foreign vaccines were available ($p$-value = 0.005). An education level below a high school diploma was significantly associated with reluctance toward vaccination because of the unavailability of the

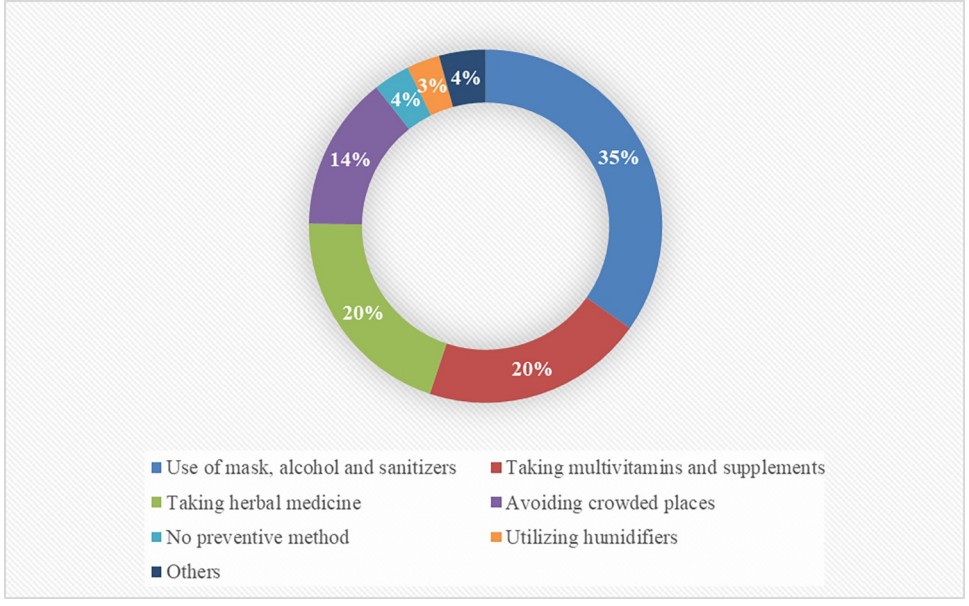

**Fig 3. Response rate to the question of "What do you do to prevent contracting COVID-19?" among the unvaccinated population in Shiraz, Iran.**

**Table 4. Association of age and gender with reluctancy towards coronavirus disease vaccination and the motivation to participate in vaccination among the general population of Shiraz.**

| Reason | Gender; n (%) | | | Age group; n (%) | | | | |
|---|---|---|---|---|---|---|---|---|
| | Male; n = 427 | Female; n = 374 | P-value* | 18–25; n = 155 | 26–45; n = 415 | 46–65; n = 178 | > 65; n = 38 | P-value* |
| **Reluctancy toward vaccination** | | | | | | | | |
| The side effects of the vaccine outweigh the benefits. | 185 (43.3) | 165 (44.1) | 0.822 | 69 (44.5) | 179 (43.1) | 80 (44.9) | 16 (42.1) | 0.970 |
| The efficacy of vaccines is unknown. | 184 (43.1) | 140 (37.4) | 0.104 | 68 (43.9) | 163 (39.3) | 75 (42.1) | 12 (31.6) | 0.486 |
| I distrust vaccine companies. | 141 (33.0) | 122 (32.6) | 0.904 | 51 (32.9) | 137 (33.0) | 58 (32.6) | 15 (39.5) | 0.869 |
| My body has a robust immune system. | 123 (28.8) | 67 (17.9) | <**0.001** | 42 (27.1) | 108 (26.0) | 33 (18.5) | 4 (10.5) | **0.037** |
| I do not need to get the vaccine because of mask and glove use. | 67 (15.7) | 38 (10.2) | **0.021** | 23 (14.8) | 59 (14.2) | 17 (9.6) | 3 (7.9) | 0.289 |
| Life is in the hands of God, and there is no need for a vaccine. | 24 (5.6) | 11 (2.9) | 0.064 | 1 (0.6) | 19 (4.6) | 12 (6.7) | 3 (7.9) | **0.036** |
| I refuse due to pregnancy or lactation. | 0 (0.0) | 34 (9.1) | <**0.001** | 8 (5.2) | 26 (6.3) | 0 (0.0) | 0 (0.0) | **0.003** |
| The vaccine I trust is not available. | 24 (5.6) | 7 (1.9) | **0.006** | 5 (3.2) | 18 (4.3) | 6 (3.4) | 0 (0.0) | 0.554 |
| I have no time to get vaccinated. | 21 (4.9) | 9 (2.4) | 0.062 | 9 (5.8) | 21 (5.1) | 0 (0.0) | 0 (0.0) | **0.007** |
| Health authorities' speech about vaccines is contradictory. | 18 (4.2) | 10 (2.7) | 0.236 | 4 (2.6) | 19 (4.6) | 5 (2.8) | 0 (0.0) | 0.337 |
| I have experienced reactions to previous vaccines. | 16 (57.1) | 12 (3.2) | 0.679 | 5 (3.2) | 15 (3.6) | 7 (3.9) | 0 (0.0) | 0.672 |
| I do not believe in the existence of COVID-19. | 22 (5.2) | 5 (1.3) | **0.003** | 2 (1.3) | 14 (3.4) | 9 (5.1) | 2 (5.3) | 0.267 |
| COVID-19 is not as intense as broadcasted. | 21 (4.9) | 6 (1.6) | **0.010** | 4 (2.6) | 19 (4.6) | 4 (2.2) | 0 (0.0) | 0.252 |
| I have already contracted COVID-19, so there is no need for a vaccination. | 16 (3.7) | 10 (2.7) | 0.392 | 3 (1.9) | 18 (4.3) | 4 (2.2) | 1 (2.6) | 0.397 |
| I trust the anti-vaxxers. | 12 (2.8) | 8 (2.1) | 0.544 | 9 (5.8) | 9 (2.2) | 1 (0.6) | 0 (0.0) | **0.022** |
| **Participation in vaccination** | | | | | | | | |
| I require assurance of the safety of the vaccine. | 169 (39.6) | 181 (48.4) | **0.012** | 63 (40.6) | 182 (43.9) | 83 (46.6) | 15 (39.5) | 0.685 |
| I require assurance of the effectiveness of the vaccine. | 136 (31.9) | 140 (37.4) | 0.097 | 58 (37.4) | 140 (33.7) | 61 (34.3) | 13 (34.2) | 0.875 |
| I will not get vaccinated under any circumstances. | 140 (32.8) | 64 (17.1) | <**0.001** | 38 (24.5) | 103 (24.8) | 47 (26.4) | 12 (31.6) | 0.802 |
| I require access to the desired vaccine. | 49 (11.5) | 41 (11.0) | 0.819 | 21 (13.5) | 59 (14.2) | 8 (4.5) | 0 (0.0) | **0.001** |
| I require secure access to approved non-domestic vaccines. | 20 (4.7) | 19 (5.1) | 0.795 | 11 (7.1) | 22 (5.3) | 5 (2.8) | 1 (2.6) | 0.285 |
| I require secure access to approved domestic vaccines. | 6 (1.4) | 13 (3.5) | 0.055 | 6 (3.9) | 10 (2.4) | 2 (1.1) | 1 (2.6) | 0.447 |
| I will vaccinate if my physician advises me thus. | 5 (1.2) | 9 (2.4) | 0.183 | 1 (0.6) | 6 (1.4) | 6 (3.4) | 1 (2.6) | 0.221 |

* Chi-square/Fisher exact test

Bold variables indicate a significant association.

desired vaccine (*p*-value = 0.041) and belief that life is in the hands of God, so vaccination is not required (*p*-value = 0.010).

Figs 4 and 5 demonstrate the reasons for reluctancy toward vaccination and factors influencing participation in vaccination based on socio-economic status in the unvaccinated population of this study.

Those participants who reported their economic status as low were significantly reluctant toward vaccination because of self-presumption of high immunity (*p*-value < 0.001), unclear efficacy of vaccines (*p*-value < 0.001), the side effects outweighing the benefits of vaccines (*p*-value < 0.001), having no trust in vaccine companies (*p*-value < 0.001), the usage of mask, gloves, and sanitizers (*p*-value < 0.001), the contradictory speech of health authorities about vaccines (*p*-value = 0.041), and the unavailability of trusted vaccines (*p*-value = 0.002). Low socioeconomic status was significantly associated with participation in vaccination programs if participants were assured of the safety (*p*-value < 0.001) and efficacy (*p*-value < 0.001) of

**Table 5. Association of educational level and social-economic status with reluctance towards coronavirus disease vaccination and motivations for participation in vaccination among the general population of Shiraz.**

| Reason | Educational level; n (%) | | | | | Socio-economic status; n (%) | | | |
|---|---|---|---|---|---|---|---|---|---|
| | Illiterate; *n = 82* | Below high school diploma; *n = 302* | High school diploma; *n = 306* | University degrees; *n = 110* | P-value[*] | Middle to high; *n = 352* | Middle to low; *n = 92* | Low; *n = 357* | P-value[*] |
| **Reluctancy toward vaccination** | | | | | | | | | |
| The side effects of the vaccine outweigh the benefits. | 29 (35.4) | 132 (43.7) | 145 (47.4) | 44 (40.0) | 0.202 | 130 (36.9) | 24 (26.1) | 196 (54.9) | **<0.001** |
| The efficacy of vaccines is unknown. | 25 (30.5) | 118 (39.1) | 133 (43.5) | 48 (43.6) | 0.156 | 109 (31.0) | 27 (29.3) | 188 (52.7) | **<0.001** |
| I mistrust vaccine companies. | 19 (23.2) | 101 (33.4) | 105 (34.3) | 38 (34.5) | 0.265 | 87 (24.7) | 10 (10.9) | 166 (46.5) | **<0.001** |
| My body has a robust immune system. | 13 (15.9) | 73 (24.2) | 78 (25.5) | 26 (23.6) | 0.339 | 61 (17.3) | 18 (19.6) | 111 (31.1) | **<0.001** |
| I do not need to get the vaccine because of mask and glove use. | 6 (7.3) | 47 (15.6) | 44 (14.4) | 8 (7.3) | 0.052 | 20 (5.7) | 4 (4.3) | 81 (22.7) | **<0.001** |
| Life is in the hands of God, and there is no need for a vaccine. | 1 (1.2) | 22 (7.3) | 11 (3.6) | 1 (0.9) | **0.010** | 11 (3.1) | 1 (1.1) | 23 (6.4) | **0.025** |
| I refuse due to pregnancy or lactation. | 2 (2.4) | 7 (2.3) | 17 (5.6) | 8 (7.3) | 0.063 | 20 (5.7) | 7 (7.6) | 7 (2.0) | **0.011** |
| The vaccine I trust is not available. | 0 (0.0) | 18 (6.0) | 11 (3.6) | 2 (1.8) | **0.041** | 8 (2.3) | 0 (0.0) | 23 (6.4) | **0.002** |
| I have no time to get vaccinated. | 7 (8.5) | 11 (3.6) | 12 (3.9) | 0 (0.0) | **0.014** | 19 (5.4) | 3 (3.3) | 8 (2.2) | 0.083 |
| Health authorities' speech about vaccines is contradictory | 1 (1.2) | 7 (2.3) | 10 (3.3) | 10 (9.1) | **0.014** | 7 (2.0) | 2 (2.2) | 19 (5.3) | **0.041** |
| I have experienced reactions to previous vaccines. | 2 (2.4) | 12 (4.0) | 9 (2.9) | 4 (3.6) | 0.861 | 8 (2.3) | 2 (2.2) | 18 (5.0) | 0.102 |
| I disbelieve in the existence of COVID-19. | 3 (3.7) | 14 (4.6) | 8 (2.6) | 2 (1.8) | 0.415 | 10 (2.8) | 4 (4.3) | 13 (3.6) | 0.721 |
| COVID-19 is not as intense as broadcasted. | 1 (1.2) | 14 (4.6) | 10 (3.3) | 2 (1.8) | 0.402 | 12 (3.4) | 3 (3.3) | 12 (3.4) | 0.997 |
| I have already contracted COVID-19, so there is no need for a vaccination. | 1 (1.2) | 7 (2.3) | 9 (2.9) | 9 (8.2) | **0.032** | 13 (3.7) | 3 (3.3) | 1 (2.8) | 0.799 |
| I trust anti-vaxxers. | 1 (1.2) | 3 (1.0) | 11 (3.6) | 5 (4.5) | 0.062 | 7 (2.0) | 6 (6.5) | 7 (2.0) | **0.031** |
| **Participation in vaccination** | | | | | | | | | |
| I require assurance of the safety of the vaccine. | 31 (37.8) | 136 (45.0) | 142 (46.4) | 41 (37.3) | 0.248 | 137 (38.9) | 23 (25.0) | 190 (53.2) | **<0.001** |
| I require assurance of the effectiveness of the vaccine. | 25 (30.5) | 98 (32.5) | 116 (37.9) | 37 (33.6) | 0.431 | 83 (23.6) | 19 (20.7) | 174 (48.7) | **<0.001** |
| I will not get vaccinated under any circumstances. | 17 (20.7) | 76 (25.5) | 80 (26.1) | 31 (28.2) | 0.686 | 95 (27.0) | 31 (33.7) | 78 (21.8) | **0.046** |
| I require access to the desired vaccine. | 9 (11.0) | 28 (9.3) | 39 (12.7) | 14 (12.7) | 0.549 | 11 (3.1) | 4 (4.3) | 75 (21.0) | **<0.001** |
| I require secure access to approved non-domestic vaccines. | 1 (1.1) | 8 (2.6) | 19 (6.2) | 11 (10.0) | **0.005** | 24 (6.8) | 3 (3.3) | 12 (3.4) | 0.076 |
| I require secure access to approved domestic vaccines. | 2 (2.4) | 4 (1.3) | 11 (3.6) | 2 (1.8) | 0.314 | 6 (1.7) | 1 (1.1) | 12 (3.4) | 0.241 |
| I will vaccinate if my physician advises me thus. | 1 (1.2) | 7 (2.3) | 2 (0.7) | 3 (2.7) | 0.304 | 10 (2.8) | 0 (0.0) | 4 (1.1) | 0.086 |

[*] Chi-square/Fisher exact test

Bold variables indicate a significant association.

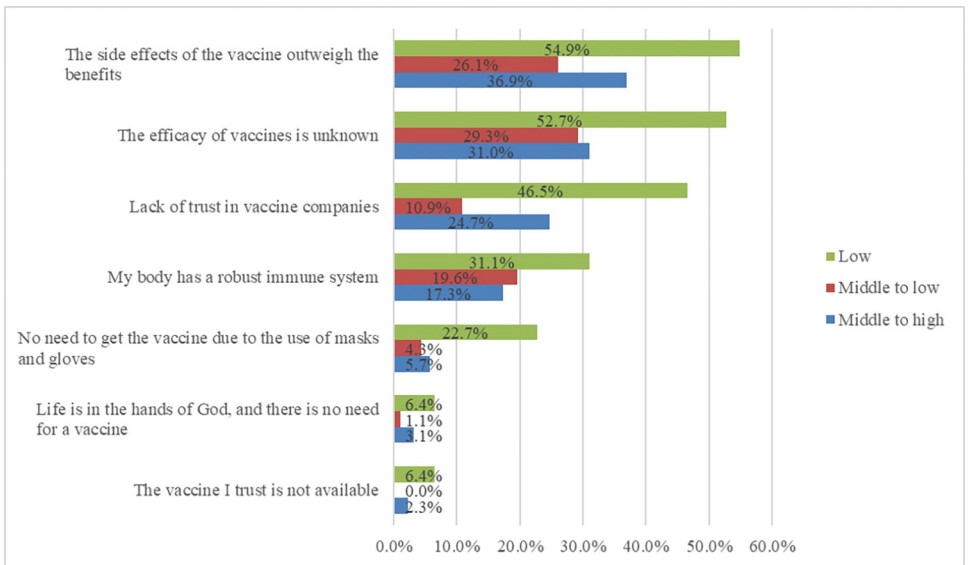

**Fig 4. Reasons for reluctancy toward vaccination based on socio-economic status among the unvaccinated population in Shiraz, Iran.**

vaccines and upon the availability of preferred ones ($p$-value < 0.001). Moreover, those who reported their socio-economic status as lower than average were more likely to never get the vaccination ($p$-value = 0.046). It should also be noted that people with low socio-economic status were more likely to obtain information about vaccine reluctance from family and friends ($p$-value <0.001) and complementary medical professionals ($p$-value <0.001).

## Discussion

The benefits of immunization undeniably outweigh the side effects, as it is one of the most influential and cost-benefit interventions in improving health status among individuals [25]. Achieving a high vaccination rate is required to reduce the morbidity and mortality in preventable diseases with immunization [25]. Keeping in mind that vaccine development for the

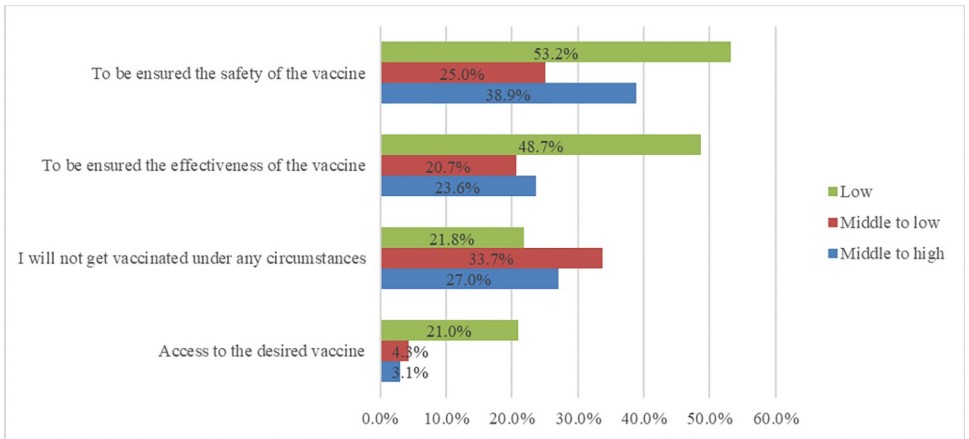

**Fig 5. Factors influencing participation in vaccination based on socio-economic status among the unvaccinated population in Shiraz, Iran.**

newly emerged disease is the first challenge to lessening its effect on public health, dealing with vaccination hesitancy and reluctance is the next battle. This phenomenon has held back the global effort toward better health, as evidenced by the re-emergence of some contagious diseases such as pertussis and measles [26, 27]. It is considered one of the top ten threats to global health in 2019 by the WHO [28]. Vaccine hesitancy is a complex problem, and there is no single intervention to prevent it; understanding the roots and causes of this phenomenon in subgroups might provide sufficient information to help authorities take necessary actions when implementing health policies.

In the current study, we evaluated the reasons for being reluctant toward vaccination, the factors that are important for individuals to participate in vaccination programs, and their associations with demographic characteristics despite their variety and availability. The results demonstrate that two significant factors prevent people from vaccinating, namely concern about vaccine efficacy and concern about side effects. Men and younger generations (18–25 years of age) tend to be more reluctant toward vaccination because of their presumption of high immunity status. Those with low socioeconomic status who primarily get their information about COVID-19 from family and friends were more reluctant toward vaccination because of concerns regarding efficacy, side effects, and not trusting the vaccine companies. Similar results have also been reported regarding low socio-economic status and age group [29, 30]. Moreover, those with university degrees were more reluctant toward vaccination because of contradictions in health authorities' speech; they were willing to participate in vaccination programs if foreign vaccines became available. Nonetheless, these factors may differ among different nations and cultures. For example, Umakanthan et al. investigated vaccine hesitancy in Germany and reported that younger age, lower education, and female gender decreased the odds of having the willingness to vaccinate [31].

Several studies have investigated the roots of vaccine hesitancy in the COVID-19 era, and most found concerns about efficacy and safety to be the most highly reported causes, similar to the present study [32–35]; however, those studies were conducted mainly in late 2020 and early 2021, when COVID-19 vaccination was not as adequately administered compared to the timeline of the current study, when Iran, similar to many other countries, was offering their citizens the fourth dose of COVID-19 vaccine. Nevertheless, the trend toward vaccine reluctance is being reduced as time passes [36, 37], and to date billions of people have been inoculated with vaccines that have proven their efficacy and safety, even toward new COVID-19 variants [38]. However, the remaining unvaccinated population is still concerned about vaccine effectiveness and security, and this situation requires increasing awareness in the public, primarily through person-to-person communications and social media, which were reported as the most frequent sources of information in the present study.

This study and others emphasize the importance of implementing policies toward a higher vaccine coverage rate equal to less morbidity and mortality and fewer adverse effects on various aspects of people's life, including social, mental, economic, and cultural dimensions. Because vaccine reluctance is a psycho-behavioral issue and may differ from one society to another based on beliefs, cultural and educational differences of populations, authorities must take the proper actions to deal with this problem reasonably, as it is not an issue related solely to COVID-19 [31]. Thus, we recommend further studies evaluating vaccine hesitancy and reluctance to be adjusted based on an organization consisting of a multidisciplinary panel of experts, especially in population hot spots [35], while periodically performing surveys in the targeted groups to understand better subgroups involved in vaccine reluctance in the present study.

Knowing that vaccine reluctance is related to ineffective communication and mass population education, this organization can learn from other countries' experiences in dealing with

vaccine hesitancy to implement policies considering the targeted sub-groups' socio-cultural beliefs and establish the method that has worked in previous battles against this issue. Policy-makers can take advantage of numerous tactics mentioned in the scientific literature while combating vaccine reluctance. For instance, the Council of Canadian Academies Expert Panel on Health Product Risk Communication Evaluation has addressed the five best practices to combat vaccine hesitancy: (I) Detect the targeted population and establish trust; (II) Provide balanced, evidence-based information regarding both the risks and benefits of being vaccinated; (III) Provide facts and address misconceptions and myths; (IV) Utilize visual-aid tools such as videos, pictures, and graphs, as they help people with little numeracy skills; (V) Test the designed communication toolkit before launching [39]. Nevertheless, it should be noted that although this method might have worked in Canada, there is no guarantee of its efficacy in Iran, as there are many socio-cultural differences between the two countries.

Some limitations faced by the current study are worth mentioning. Only individuals above 18 years of age were included, so there is no data regarding the hesitancy of those under 18 years of age or the attitude of their parents toward vaccination. Moreover, the influence of available treatments and the availability and company of vaccines were not discussed, as many individuals were reluctant toward vaccines because of the unavailability of desired ones, such as Pfizer-BioNTech, Moderna, and Johnson and Johnson, among others. Furthermore, as this study was based on a self-reporting questionnaire, it is susceptible to some biases, such as recall bias, that might affect the outcome of the results.

## Conclusion

As demonstrated in the current study, avoiding vaccination is an undeniable public and individual health concern in Iran. Worrying about vaccine efficacy and side effects are the most reported causes of vaccine reluctance among individuals, which could be altered by emphasizing mass education and averting an infodemic by forming dedicated multidisciplinary organizations. Therefore, health authorities must take action and combat vaccine reluctance to increase vaccination awareness, especially among vulnerable groups.

## Acknowledgments

This study was the subject of the MPH degree thesis for Parisa Hossseinpour and Amirhossein Erfani.

## Author Contributions

**Conceptualization:** Najmeh Maharlouei, Kamran Bagheri Lankarani.

**Data curation:** Amirhossein Erfani, Reza Shahriarirad, Hadi Raeisi Shahrakie, Abbas Rezaianzadeh.

**Formal analysis:** Amirhossein Erfani, Reza Shahriarirad.

**Funding acquisition:** Kamran Bagheri Lankarani.

**Methodology:** Najmeh Maharlouei, Parisa Hosseinpour.

**Project administration:** Abbas Rezaianzadeh, Kamran Bagheri Lankarani.

**Supervision:** Najmeh Maharlouei, Parisa Hosseinpour, Amirhossein Erfani, Kamran Bagheri Lankarani.

**Validation:** Najmeh Maharlouei.

**Writing – original draft:** Parisa Hosseinpour, Amirhossein Erfani.

**Writing – review & editing:** Najmeh Maharlouei, Parisa Hosseinpour, Amirhossein Erfani, Reza Shahriarirad, Hadi Raeisi Shahrakie, Abbas Rezaianzadeh, Kamran Bagheri Lankarani.

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
