## [Decision Letter · Decision Letter 0]

5 Oct 2022

PONE-D-22-26777Associated Factors of Reluctancy Toward COVID-19 Vaccination; A Cross-Sectional Study in Shiraz, Southern Iran PLOS ONE

Dear Dr. Amirhossein Erfani,

Thank you for submitting your manuscript to PLOS ONE. After careful consideration, we feel that it has merit but does not fully meet PLOS ONE’s publication criteria as it currently stands. Therefore, we invite you to submit a revised version of the manuscript that addresses the points raised during the review process.

Please submit your revised manuscript by Nov 19 2022 11:59PM. If you will need more time than this to complete your revisions, please reply to this message or contact the journal office at plosone@plos.org. Please include the following items when submitting your revised manuscript:A rebuttal letter that responds to each point raised by the academic editor and reviewer(s). You should upload this letter as a separate file labeled 'Response to Reviewers'.A marked-up copy of your manuscript that highlights changes made to the original version. You should upload this as a separate file labeled 'Revised Manuscript with Track Changes'.An unmarked version of your revised paper without tracked changes. You should upload this as a separate file labeled 'Manuscript'.If applicable, we recommend that you deposit your laboratory protocols in protocols.io to enhance the reproducibility of your results. Protocols.io assigns your protocol its own identifier (DOI) so that it can be cited independently in the future. For instructions see: https://journals.plos.org/plosone/s/submission-guidelines#loc-laboratory-protocols. Additionally, PLOS ONE offers an option for publishing peer-reviewed Lab Protocol articles, which describe protocols hosted on protocols.io. Read more information on sharing protocols at https://plos.org/protocols?utm_medium=editorial-email&utm_source=authorletters&utm_campaign=protocols.

We look forward to receiving your revised manuscript.

Kind regards,

Srikanth Umakanthan

Academic Editor

PLOS ONE

Journal Requirements:

 Whilst you may use any professional scientific editing service of your choice, PLOS has partnered with both American Journal Experts (AJE) and Editage to provide discounted services to PLOS authors. Both organizations have experience helping authors meet PLOS guidelines and can provide language editing, translation, manuscript formatting, and figure formatting to ensure your manuscript meets our submission guidelines. To take advantage of our partnership with AJE, visit the AJE website (http://aje.com/go/plos) for a 15% discount off AJE services. To take advantage of our partnership with Editage, visit the Editage website (www.editage.com) and enter referral code PLOSEDIT for a 15% discount off Editage services. If the PLOS editorial team finds any language issues in text that either AJE or Editage has edited, the service provider will re-edit the text for free.

"This study was the subject of the MPH degree thesis for Parisa Hossseinpour and Amirhossein Erfani. The authors would like to thank the Vice Chancellor for Research of Shiraz University of Medical Sciences for financially supporting the project (Project code: 22574)."

"Vice-chancellor for Research of Shiraz University of Medical Sciences financially supported this Study through Kamran Bagheri Lankarani (Grant No: 24974)."

5. Please include correct caption for figures.

Additional Editor Comments:

The manuscript requires minor revisions as stated by the reviewers. Include a Point-to-point inclusion of the suggestions/comments to improvise the manuscript.

Reviewers' comments:

Reviewer's Responses to Questions

**Comments to the Author**

1. Is the manuscript technically sound, and do the data support the conclusions?

Reviewer #1: Yes

Reviewer #2: Yes

2. Has the statistical analysis been performed appropriately and rigorously? 

Reviewer #1: Yes

Reviewer #2: Yes

3. Have the authors made all data underlying the findings in their manuscript fully available?

Reviewer #1: Yes

Reviewer #2: Yes

4. Is the manuscript presented in an intelligible fashion and written in standard English?

Reviewer #1: Yes

Reviewer #2: Yes

5. Review Comments to the Author

Reviewer #1: Well written manuscript that reflects the Associated Factors of Reluctancy Toward COVID-19 Vaccination in Iran.

The manuscript can be strengthened by incorporating the following points:

1. Include a short note on the origin of COVID-19 (refer and cite: doi: 10.1136/postgradmedj-2020-138234

2. Compare the COVID-19 states in Iran with other regions (refer and cite: doi: 10.3389/fpubh.2022.844333)

3. The role of vaccination status that has declined the vaccine resistance rates(refer and cite: doi: 10.3390/vaccines9101064.)

4. How the Iranian government and health care has imbibed regulations to combat COVID-19 targeting its predictors (refer and cite: doi: 10.1136/postgradmedj-2021-141365)

5. The treatment of COVID-19 and its implications on the vaccine hesitancy (refer and cite: doi: 10.3389/fphar.2022.742273.

6. Include other forms of representations in your figures (eg. Bar charts, histograms), color the images for better viewership.

Reviewer #2: The authors have finely incorporated the Associated Factors of Reluctancy Toward COVID-19 Vaccination in Iran.

Grammatical errors need to be corrected.

Include illustrations or bar charts. The introduction is very verbose. Needs to be tapered.

6. PLOS authors have the option to publish the peer review history of their article (what does this mean?). If published, this will include your full peer review and any attached files.

Reviewer #1: No

Reviewer #2: No

---

## [Author Response · Author response to Decision Letter 0]

23 Nov 2022

Journal Requirements:

Authors Response: We apologize for this inconvenience and have revised the manuscript based on the provided guidelines.

Authors Response: Our manuscript has been revised by a Native English editor and we have attached the certificate for your reference.

Authors Response: We have uploaded the mentioned files as requested.

"This study was the subject of the MPH degree thesis for Parisa Hossseinpour and Amirhossein Erfani. The authors would like to thank the Vice Chancellor for Research of Shiraz University of Medical Sciences for financially supporting the project (Project code: 22574)."

We note that you have provided funding information that is not currently declared in your Funding Statement. However, funding information should not appear in the Acknowledgments section or other areas of your manuscript. We will only publish funding information present in the Funding Statement section of the online submission form. Please remove any funding-related text from the manuscript and let us know how you would like to update your Funding Statement. Currently, your Funding Statement reads as follows: "Vice-chancellor for Research of Shiraz University of Medical Sciences financially supported this Study through Kamran Bagheri Lankarani (Grant No: 24974)."

Authors Response: We have moved the funding statement to the cover letter as requested.

Authors Response: We have moved the ethical consideration section to the method and material section as requested.

6. Please include correct caption for figures.

Authors Response: We have adjusted and corrected the figure captions based on the journal’s guidelines.

Authors Response: We have revised the manuscript reference list based on the journal’s guidelines.

Comments to the Author

Dear Editor and Reviewers,

Thanks for reaching out to us regarding the manuscript entitled “Associated Factors of Reluctancy Toward COVID-19 Vaccination; A Cross-Sectional Study in Shiraz, Southern Iran ". We believe that these comments have helped us enhance the quality of the manuscript. We also have done our best to revise and improve the paper according to the comments. Herewith, we provided the authors' responses to each comment right after each statement. Also, all the changes have been addressed in the manuscript through highlighted parts according to journal policies.

• Reviewer #1: 

1. Include a short note on the origin of COVID-19 (refer and cite: doi: 10.1136/postgradmedj-2020-138234

Authors’ Response: Thank you for your comment. A statement regarding the origin of COVID-19 has been added to the manuscript according to the provided reference. (Lines 44 – 51)

2. Compare the COVID-19 states in Iran with other regions (refer and cite: doi: 10.3389/fpubh.2022.844333)

Author’s Response: Thank you for your comment and for providing us with this valuable article. A statement has been added to the manuscript addressing the economic effects of COVID-19. (Lines 47 – 51)

3. The role of vaccination status that has declined the vaccine resistance rates(refer and cite: doi: 10.3390/vaccines9101064.)

Authors’ Response: Thank you for your comment. This article was referred to in the discussion section regarding the concerns for the post-vaccine scare of adverse health effects. (Discussion; lines 266 – 268)

4. How the Iranian government and health care has imbibed regulations to combat COVID-19 targeting its predictors (refer and cite: doi: 10.1136/postgradmedj-2021-141365)

Authors’ Response: Thank you for the valuable comment. Since combating vaccine hesitancy might differ from one society to another, a statement regarding considering cultural beliefs and educational status was added to the manuscript. (Paragraph 2 Discussion section; lines 263 – 265; and Paragraph 4 discussion section; line 281 – 283)

5. The treatment of COVID-19 and its implications on the vaccine hesitancy (refer and cite: doi: 10.3389/fphar.2022.742273.

Authors’ Response: Thank you for your comment. Although we did not evaluate the effect of treatment on vaccine hesitancy, we have added this statement in the limitation section and also a statement regarding the practice of available medicines in combating the COVID-19 pandemic has been added to the manuscript (Lines 52 and 53).

6. Include other forms of representations in your figures (eg. Bar charts, histograms), color the images for better viewership.

Authors’ Response: Thank you for your comment. Two more figures regarding the reasons of reluctancy toward vaccination and influencing the of participation in vaccination based on socio-economic status has been added to the manuscript (Figure 4 and 5).

• Reviewer #2: 

1. Grammatical errors need to be corrected.

Authors’ Response: Thank you for your valuable comment. The manuscript has been revised in aspect of grammatical errors by a professional English editor.

2. Include illustrations or bar charts. 

Authors’ Response: Thank you for your comment. Two more figures regarding the reasons of reluctancy toward vaccination and influencing the of participation in vaccination based on socio-economic status has been added to the manuscript (Figure 4 and 5).

3. The introduction is very verbose. Needs to be tapered.

Authors’ Response: Thank you for your comment and concern. Based on the first reviewers comments we were obligated to add some additional requested information. Furthermore, we believe that any tapering in the information provided in the introduction section will interrupt the continuity and understanding of the phases and line of thought for the general readers. However, if the honorable reviewer believes that a certain part of the introduction is abundant, we would gladly revise the mentioned section accordingly.

---

## [Editor Report · Decision Letter 1]

25 Nov 2022

Factors associated with reluctancy to acquire COVID-19 vaccination: a cross-sectional study in Shiraz, Iran, 2022 

PONE-D-22-26777R1

Dear Dr. Erfani,

We’re pleased to inform you that your manuscript has been judged scientifically suitable for publication and will be formally accepted for publication once it meets all outstanding technical requirements.

Kind regards,

Srikanth Umakanthan

Academic Editor

PLOS ONE

Additional Editor Comments (optional):

Accept in revised format
---

## [Editor Report · Acceptance letter]

2 Dec 2022

PONE-D-22-26777R1 

Factors associated with reluctancy to acquire COVID-19 vaccination: a cross-sectional study in Shiraz,Iran, 2022  

Dear Dr. Erfani:

I'm pleased to inform you that your manuscript has been deemed suitable for publication in PLOS ONE. Congratulations! Your manuscript is now with our production department. 

Kind regards, 

on behalf of

Dr. Srikanth Umakanthan 

Academic Editor

PLOS ONE